# OpenReview forum: "Test-Time Safety Alignment with Dynamic Intervention for Jailbreak Defense in LLMs"
_ICLR.cc/2026/Conference — ICLR 2026 Conference Withdrawn Submission_

### Official Review · Reviewer_vnDo · 2025-10-28

**Soundness:** 1
**Presentation:** 1
**Contribution:** 3
**Rating:** 2
**Confidence:** 4

**Summary:**

This paper proposes TRADE (Test-time Security Alignment with Dynamic Intervention), a framework that defends large language models against jailbreak attacks during inference without retraining. TRADE integrates a reward-guided branch update for efficient token evaluation, a safety-aware post-processing step that detects harmful outputs and triggers controlled regeneration with safe prompts, and an adaptive tree search thresholding mechanism to prune unsafe branches efficiently.

**Strengths:**

1. The paper explores the critical issue of ensuring LLM safety against jailbreak attacks, focusing on inference-time defense, which is more lightweight and practical for real-world deployment.
2. The proposed TRADE framework is interesting, introducing a new way to achieve test-time safety alignment through dynamic intervention, which distinguishes it from existing static or post-hoc defense methods.
3. The experiments provide convincing results demonstrating robustness under parameter changes and stable general reasoning performance, supporting the effectiveness and reliability of the approach.

**Weaknesses:**

1. The paper claims to “investigate the security of large language models,” but the scope of “security” is much broader, encompassing jailbreak, prompt injection, privacy leakage, and etc. But it seems like the proposed method only mitigates jailbreak attacks. Can authors demonstrate with evidence that the method generalizes to other types of security threats? If not, it would be more appropriate to narrow the scope and clearly frame the work as focusing on jailbreak defense.

2. If the actual research scope of this paper is jailbreak defense, the introduction and related work should clearly focus on this direction. However, the paper still discusses the topic in broad terms such as “security” and “safety,” and the proposed method is motivated by the limitations of general safety-alignment approaches rather than prior jailbreak-specific methods. This leads to a mismatch between the claimed motivation and the actual contribution. Could the authors clarify why existing jailbreak defense studies (e.g., [1–2]) were not taken into consideration in the discussion or comparison?

3. Jailbreak attacks are rapidly evolving, including cipher-based methods [3], multi-turn jailbreaks [4], and other adaptive attacks [5]. How does the proposed method perform when faced with these attacks?

4. While TRADE claims to be computationally efficient, the paper does not provide clear quantitative comparisons of runtime or latency overhead relative to other inference-time defense methods. Could the authors provide empirical evidence or measurements to support the claimed efficiency?

5. The paper presents theoretical analyses, but these results are not empirically linked to the observed experimental behavior. There is no quantitative or qualitative evidence showing how the theoretical bounds correspond to actual performance gains in jailbreak defense. Could the authors clarify how the proposed theoretical results are reflected in or supported by the empirical findings?

References.
[1]. Xie, Y., Yi, J., Shao, J., Curl, J., Lyu, L., Chen, Q., ... & Wu, F. (2023). Defending chatgpt against jailbreak attack via self-reminders. Nature Machine Intelligence, 5(12), 1486-1496.

[2]. Zhou A, Li B, Wang H. Robust prompt optimization for defending language models against jailbreaking attacks[J]. Advances in Neural Information Processing Systems, 2024, 37: 40184-40211.

[3]. Yuan, Y., Jiao, W., Wang, W., Huang, J. T., He, P., Shi, S., & Tu, Z. (2023). Gpt-4 is too smart to be safe: Stealthy chat with llms via cipher. arXiv preprint arXiv:2308.06463.

[4]. Russinovich, M., Salem, A., & Eldan, R. (2025). Great, now write an article about that: The crescendo {Multi-Turn}{LLM} jailbreak attack. In 34th USENIX Security Symposium (USENIX Security 25) (pp. 2421-2440).

[5]. Kuo, M., Zhang, J., Ding, A., Wang, Q., DiValentin, L., Bao, Y., ... & Chen, Y. (2025). H-cot: Hijacking the chain-of-thought safety reasoning mechanism to jailbreak large reasoning models, including openai o1/o3, deepseek-r1, and gemini 2.0 flash thinking. arXiv preprint arXiv:2502.12893.

**Questions:**

See my aforementioned weakness.

---

### Official Review · Reviewer_e2Py · 2025-10-28

**Soundness:** 2
**Presentation:** 2
**Contribution:** 2
**Rating:** 4
**Confidence:** 3

**Summary:**

This paper proposes TRADE, a framework for defending large language models against jailbreak attacks during inference. The approach combines three key components: (1) a reward-guided branch update module using a multifurcation reward model to efficiently evaluate multiple candidate tokens simultaneously, (2) safety-aware post-processing that detects harmful outputs and triggers regeneration with secure prompts, and (3) an efficient tree search thresholding strategy with adaptive pruning based on information gain. The authors provide theoretical analysis showing logarithmic marginal performance decay and adaptive safety threshold convergence, and conduct experiments on HarmfulHExPHI, AdvBench, and JBB-Behaviors datasets.

**Strengths:**

1. The paper addresses an important gap by focusing on inference-time jailbreak defense rather than training-time alignment. The theoretical analysis provides rigorous justification for the logarithmic decay of marginal performance gains and the adaptive threshold design, establishing a principled basis for the pruning strategy that balances safety and computational efficiency.

2. TRADE integrates three well-designed modules that work synergistically—the multifurcation reward model improves efficiency by evaluating multiple tokens in parallel, the safety-aware post-processing provides an additional layer of defense with external verification, and the adaptive tree search prevents computational explosion during regeneration. The ablation studies clearly demonstrate that each component contributes meaningfully to the overall defense performance.

**Weaknesses:**

- Limited computational cost analysis: While efficiency is claimed as a key advantage, the paper lacks detailed analysis of inference latency, memory overhead, and throughput comparisons. The actual computational cost relative to standard decoding and baseline methods is not quantified.

- Unclear generalization across architectures: The experiments focus narrowly on Llama and Mistral model families with Llama Guard as the safety classifier. The transferability to other model architectures and reward models remains unexplored.

- Weak adaptive attack evaluation: The baselines tested (Best-of-N, MCTS, Self-Consistency) are not adversarial attacks. The robustness against attacks specifically designed to exploit TRADE's mechanisms is not demonstrated.

**Questions:**

- Can the authors evaluate against sophisticated adaptive attacks? Specifically, how does TRADE perform against optimization-based jailbreaks like GCG, GPTFuzzer, AutoDAN, PAIR, or other attacks that have proven effective against aligned models?

- What are the concrete computational costs? Can the authors provide quantitative metrics including: inference latency (tokens/second), wall-clock time per query, memory usage, and total computational overhead as a function of N and L? How do these costs compare to standard decoding and each baseline method?

- How well does TRADE generalize across different model families and safety classifiers? Does the approach work with other architectures (GPT-style, Claude, Gemini)? Do the hyperparameters require extensive re-tuning when changing the base model or reward model, or do they transfer effectively?

---

### Official Review · Reviewer_bbAG · 2025-11-01

**Soundness:** 2
**Presentation:** 3
**Contribution:** 2
**Rating:** 2
**Confidence:** 4

**Summary:**

The paper provides a safety algorithm at inference time for decoding that deploys a reward model to guide the generation. Then, they argue that their tree search algorithm further improves results. They also do restarts every time that the guardrail model detects an unsafe output.

**Strengths:**

1- While being widely popular in terms of performance in the literature, decoding strategies are less established in safety and this paper investigates their benefits.

2- The method is well explained and all the elements are clear.

**Weaknesses:**

1- The paper uses some unnecessary notation that would be better off if not used. For instance, the explanation for top-p could be in text and equations (1) and (2) do not have any points. I think authors have put so many equations overall.

2- As I understand from equation (4), the method deploys beam-search which is well studied in the literature. Then they talk about some pruning threshold that has some parameter gamma  = 1. What is the intuition for this use? Are they using a fancier version of beam-search and why? What is the point of theorems in terms of the performance?

3- I don't view the Post-processing part as a novelty of this paper. This is just best-of-N using llama-guard to choose the best response. Are the authors using the same llama-guard model in BoN as the baseline?

4- As I understand all the comparisons on the three datasets is for direct malicious queries and in absence of any jailbreaking. While I don't necessarily see that as a weakness, I would like to see how the method performs on SOTA jailbreaking methods (I will potentially increase my score based on the quality of these results)

5- To me, the ablation studies in Table 3 is not convincing that for a reasonable value of N, the tree search is crucial to the performance. For instance, for N=8 and on AdnBench the numbers are 0.1846 and 0.1993. I am pessimistic about the precision of these numbers; what is the standard deviation in these cases?

**Questions:**

Pls see above for my questions. Also:

If I understand correctly, you use the restarting trick in your algorithm while other baselines except for BoN do not. I think this setting is not fair especially because you're using llama-guard for this. Can you please justify the setting?

---

### Official Review · Reviewer_nUoA · 2025-11-02

**Soundness:** 3
**Presentation:** 3
**Contribution:** 2
**Rating:** 2
**Confidence:** 4

**Summary:**

The paper proposes TRADE, a test-time jailbreak defense that steers generation via multi-branch search, token-level reward scoring, and a post-generation safety check with restart logic. It avoids retraining the base model and emphasizes balancing search efficiency with safety by pruning unsafe branches and regenerating when outputs fail a guardrail check. Experiments on jailbreak benchmarks with 1B–8B models show lower attack success rates than standard decoding and prior inference-time reasoning methods.

**Strengths:**

- the paper is written in a very intuitive way, introducing a clear test-time intervention pipeline (branch search + safeguard + restart).

- in addition to the defensive method, additional efforts are conducted to improve the efficiency of the algorithm.

**Weaknesses:**

- The paper lacks comparison to stronger baselines such as smoothLLM, which is also a test-time inference method [1]. The empirical studies are seemingly only compare to methods that do not offer explicit defensive measure, large portion of related works have been ignored.

- The model evaluated seem to be too small (1B-8B), it is not necesaarily representative of behaviors of the larger models.

- There exist jailbreak methods that specifically fool guardrail models that check outputs after generation, such as [2], and similar strategies may also bypass the token-level safety checks used here. The paper does not evaluate against these adaptive jailbreak techniques.

- The theory focuses on generic tree-search error decay and threshold convergence rather than adversarial behavior or jailbreak failure bounds. It does not provide theoretical support for safety robustness.


[1] SmoothLLM: Defending Large Language Models Against Jailbreaking Attacks.

[2] Jailbreaking Large Language Models Against Moderation Guardrails via Cipher Characters

**Questions:**

please address the first three points in the weakness.

**Details Of Ethics Concerns:**

The paper is about defending jailbreak, the broader topic discussed is a highly controversial topic, but since the discussion is on the defending side, it's probably fine.

---

### Note · Authors · 2025-11-22

I have read and agree with the venue's withdrawal policy on behalf of myself and my co-authors.